# A Comparison of the Impact of Restrictive Diets on the Gastrointestinal Tract of Mice

**DOI:** 10.3390/nu14153120

**Published:** 2022-07-29

**Authors:** András Gregor, Laura Huber, Sandra Auernigg-Haselmaier, Felix Sternberg, Magdalena Billerhart, Andreas Dunkel, Veronika Somoza, Manfred Ogris, Barbara Kofler, Valter D. Longo, Jürgen König, Kalina Duszka

**Affiliations:** 1Department of Nutritional Sciences, University of Vienna, 1090 Vienna, Austria; andras.gregor@univie.ac.at (A.G.); lahula.lh@gmail.com (L.H.); sandra.haselmaier@univie.ac.at (S.A.-H.); juergen.koenig@univie.ac.at (J.K.); 2Department of Biomedical Sciences, Institute of Physiology, Pathophysiology and Biophysics, University of Veterinary Medicine, 1210 Vienna, Austria; felix.sternberg@vetmeduni.ac.at; 3Laboratory of Macromolecular Cancer Therapeutics (MMCT), Department of Pharmaceutical Sciences, University of Vienna, 1090 Vienna, Austria; magdalena.billerhart@boku.ac.at (M.B.); m.ogris@univie.ac.at (M.O.); 4Leibniz Institute for Food Systems Biology at the Technical University of Munich, 85354 Freising, Germany; a.dunkel.leibniz-lsb@tum.de (A.D.); v.somoza.leibniz-lsb@tum.de (V.S.); 5Research Program for Receptor Biochemistry and Tumor Metabolism, Department of Pediatrics, University Hospital of the Paracelsus Medical University, 5020 Salzburg, Austria; b.kofler@salk.at; 6Longevity Institute, Leonard Davis, Los Angeles, CA 90089, USA; vlongo@usc.edu

**Keywords:** gastrointestinal tract, caloric restriction, fasting

## Abstract

The rate of gut inflammatory diseases is growing in modern society. Previously, we showed that caloric restriction (CR) shapes gut microbiota composition and diminishes the expression of inflammatory factors along the gastrointestinal (GI) tract. The current project aimed to assess whether prominent dietary restrictive approaches, including intermittent fasting (IF), fasting-mimicking diet (FMD), and ketogenic diet (KD) have a similar effect as CR. We sought to verify which of the restrictive dietary approaches is the most potent and if the molecular pathways responsible for the impact of the diets overlap. We characterized the impact of the diets in the context of several dietary restriction-related parameters, including immune status in the GI tract; microbiota and its metabolites; bile acids (BAs); gut morphology; as well as autophagy-, mitochondria-, and energy restriction-related parameters. The effects of the various diets are very similar, particularly between CR, IF, and FMD. The occurrence of a 50 kDa truncated form of occludin, the composition of the microbiota, and BAs distinguished KD from the other diets. Based on the results, we were able to provide a comprehensive picture of the impact of restrictive diets on the gut, indicating that restrictive protocols aimed at improving gut health may be interchangeable.

## 1. Introduction

The rate of diagnosed diseases and malfunctions associated with intestinal inflammation, including inflammatory bowel disease (IBD), irritable bowel syndrome (IBS), as well as food allergies and intolerances, is growing in modern societies [1,2]. A Western diet increases the risk of developing IBD, while several dietary interventions have been shown to affect inflammatory pathways, leading to improvement and relief of the symptoms [3]. Diet may contribute to the development of intestinal inflammation through several mechanisms, including dietary antigen presentation, alterations in the mucosal immune system, epithelial barrier function, or gut microbiome [4]. On the other hand, dietary interventions are the most efficient and most commonly applied ways of improving health. Various diet plans can involve the modification of the type and amount of nutrients, but also the frequency and timing of meals. Fasting and other restrictive dietary regimens exert overall beneficial effects on health, diminish general inflammation, and gut permeability [5,6]. However, restrictive diets include a wide range of dietary protocols. Caloric restriction (CR) involves the reduction of total average daily energy intake without malnutrition and is one of the primary intervention tools for weight loss and health maintenance. CR has been repeatedly reported to increase life and health span in multiple species [7]. In the gastrointestinal (GI) tract, CR reduces cell proliferation in the duodenum and colorectal crypt cells, as well as enterocyte differentiation; influences innate and adaptive immunity in the intestine; and counteracts the age-related decline in nutrient absorption [8,9,10]. We previously reported that CR stimulates metabolic genes expression and reduces inflammatory factors’ expression along the GI tract [11]. Another restrictive regiment, intermittent fasting (IF), implicates temporal fasting with adequate caloric intake. It has been shown to extend lifespan in mice [12] and reduce symptoms of IBS as well as intestinal and colonic inflammation [6,13]. Next, the fasting-mimicking diet (FMD) is a low-calorie 4-day fasting protocol repeated in cycles. FMD aims to deliver 10–50% of the required daily calories in the form of highly nutritious meals for a restricted period. FMD has been demonstrated to improve symptoms and pathogenesis of IBD. It reduces intestinal inflammation, increases stem cell number, stimulates protective gut microbiota, as well as reverses intestinal pathology caused by DSS, including infiltration of lymphocytes in the colon’s crypt as well as accumulation of CD4^+^ cells in the colon and small intestine [14,15]. FMD is even more efficient in treating intestinal inflammation than fasting [14]. FMD has recently been further researched in the context of diabetes [16], breast cancer [17], Parkinson’s disease [18], and multiple sclerosis [19]. The final restrictive protocol of interest for the current study, the ketogenic diet (KD), gained great recognition due to its impact on cognitive performance, overall health benefits, application in cancer therapy, and its plausible anti-inflammatory properties [20,21]. KD is classically employed in the therapy of epilepsies but has also found application in other neurogenerative diseases, including amyotrophic lateral sclerosis, Alzheimer’s, and Parkinson’s disease [20]. KD has recently been re-discovered by the scientific as well as nutritionist community. Due to its prevalence, many claims have been assigned to KD; however, there is still very limited evidence verifying the outcomes of the diet. Contrary to the previously mentioned restrictive diets, KD is consumed *ad libitum*, and therefore, it does not involve caloric or time restriction but limits macronutrients. Typically, KD, with its high fat, moderate protein, and very low carbohydrate intake leads to a fasting-like ketogenic state. Despite the anti-inflammatory claims concerning restrictive diets, the molecular mechanism, as well as the impact on the GI tract, remains largely unresolved.

In a previous study [11], we showed that CR decreases the expression of inflammatory and antimicrobial factors in the mucosa along the whole GI tract of mice, suggesting a reduction of inflammation. We also demonstrated the impact of CR on microbiota composition, bile acids (BA) secretion, and short-chain fatty acids (SCFA) production [11,22,23]. In the current project, we compare selected restrictive dietary protocols CR, IF, FMD, and KD in mice. We investigated which of the protocols have similar beneficial outcomes to CR and aimed at identifying which diet shows the strongest potential in maintaining a healthy gut. We verified the impact of restrictive diets on the expression of anti-inflammatory and antimicrobial factors and researched if intestine and colon morphology; mucus production; microbiota composition; levels of SCFAs and BAs; as well as markers of autophagy, mitochondria, oxidative stress, and CR in the GI tract are affected by the diets. Importantly, multiple studies assess the impact of diets against diseases or inflammatory conditions to prove therapeutic applicability. In contrast, we aimed at characterizing the impact of the diets on the healthy intestine, expecting more subtle phenotype changes and likely, less pronounced differences between the dietary protocols than in disease states. However, this approach enables identifying diets improving GI tract functions and preventing diseases in overall healthy individuals.

## 2. Materials and Methods

### 2.1. Animal Experiments

The C57BL/6 wild-type male mice were purchased from the Jackson Laboratory. The animals were housed in standard specific-pathogen-free conditions with a 12/12 h light/dark cycle and controlled temperature and humidity. Mice were randomly divided into experimental groups of eight animals, each as follows: *ad libitum* chow-fed control mice (Ad lib); ketogenic diet (KD) *ad libitum* for 4 weeks (80% energy from fat, 15% energy from protein, and 5% from carbohydrates; diet composition as previously published [24]); CR mice submitted to 14 days constant restriction with 80% of regular *ad libitum* chow intake as previously applied in our experiments [11,23]; IF group with mice fasted every other day for 24 h (100% restriction) with a 24 h chow *ad libitum* refeeding period for a total of 28 days; FMD group submitted to three cycles of 4 days of fasting (50% kcal of *ad libitum* intake on day one and 10% kcal on days 2–4) with 7 days of refeeding as described before [5]. The FMD diet consists of vitamin and mineral-rich broth, vegetable mix, and oils blended in a low-caloric paste and dosed daily. The experimental setup aimed at exposing mice to an approximately similar level of restriction. Therefore, the strictest restriction was applied for the shortest time. The experiments were initiated so that the animals were age-matched (16 weeks old) on the dissection day. Therefore, the treatment was started at ages 12, 14, 12, and 13 for KD, CR, IF, and FMD, respectively. All eight animals per group that started the treatment survived until the end of the experiment. The animals were sacrificed by anesthesia overdose followed by cardiac puncture to draw blood. The collected tissues were snap frozen in liquid nitrogen followed by storage at −80 °C. All animal experimentation protocols were approved by the Bundesministerium für Wissenschaft, Forschung und Wirtschaft, Referat für Tierversuche und Gentechnik (BMBWF-66.006/0008-V/3b/2018). All experiments were carried out according to animal experimentation Animal Welfare Act guidelines.

### 2.2. qRT-PCR

The extraction and mRNA expression analysis were performed as previously described [22]. For mtDNA levels quantification, DNA was extracted from intestinal mucosa samples using QIAamp Fast DNA Tissue Kit (Qiagen, Germany). The quantification was performed as described previously [25]. Bacteria DNA levels were assessed by qRT-PCR following fecal DNA extraction (innuPREP Stool DNA Kit, Analytik Jena, Germany). The relative levels of bacteria abundance were calculated with 16S RNA CT value as a reference. All primers’ sequences are presented in Appendix A.

### 2.3. Western Blot

Small intestine mucosa tissue (10–15 µg) was lysed in RIPA buffer (10×, EMD Millipore Billerica, MA, USA) with Pierce Phosphatase Inhibitor (Mini Tablets, Thermo Scientific, Rockford, IL, USA) and Protease inhibitor cocktail (cOmplete Mini, EDTA-free, Roche, Mannheim, Germany). The extracted proteins were quantified using the PierceTM BCA protein determination method (Thermo Scientific). The protein concentration was 1 µg/µL for β-actin, and 3 µg/µL for IkBα and Phospho-IkBα. A total of 10 µL of each sample mix was loaded on the gel. Precision Plus ProteinTM (Bio Rad, Carlsbad, CA, USA) standard served as a molecular weight standard. The separated proteins were electro-blotted onto PVDF membranes (Bio Rad, Hercules, CA, USA) using the Trans-Blot^®^ Turbo Transfer System (Bio Rad). The membranes were dried for 1 hour, blocked, and incubated with primary antibody (dilution 1:1000) against Β-Actin (#4970; 1:1000 in 2.5% BSA TBST), IkBα (#9242; 1:1000 in 5% milk powder TBST), occludin (#91131; 1:500 in 2.5% milk powder TBST), LC3 (#12741; 1:1000 in 5% BSA TBST), ATG7 (#8558; 1:1000 in 5% BSA TBST), and AKT (#9271; 1:1000 in 5% BSA TBST) (all from Cell Signaling Technology, Danvers, MA, USA) overnight at 4 °C. Afterwards, the membranes were washed and incubated with corresponding secondary antibodies (anti-mouse IgG (#7076) or anti-rabbit IgG (#7074) dilution 1:3000; Cell Signaling Technology, Danvers, MA, USA)). Visualization of the bands was performed using SuperSignal West Dura solution (Thermo Scientific) and ChemiDoc XRS+ (Bio-Rad). Quantitative analysis of the expression was carried out with ImageJ software (version 1.53, NIH, Bethesda, MD, USA). The relative expression of the protein was calculated based on β-Actin as reference. Eight biological replicate samples per experimental group on separate Western blot lanes were used to measure each protein expression.

### 2.4. Protein Array

The samples of small intestine mucosa (10–20 µg) were disrupted in 100 µL 1× RIPA lysis buffer with PMSF (Sigma Aldrich, St. Louis, MO, USA). The solution was centrifuged for 20 min, at 10,000× *g*, at 4 °C, and protein concentration in the supernatant was determined (Pierce BCA Protein Assay Kit, Thermo Scientific). Samples were pooled to create 3–4 replicates per experimental group and 100 µL of the solution with 1 µg/µL protein was applied on the Quantibody^®^ Mouse TH17 Array 1 (RayBiotech, Peachtree Corners, GA, USA). The arrays were processed according to the manufacturer’s instructions.

### 2.5. Quantification of NFκB

The quantification of NFκB and (Ser536) p-NFκB was performed using ELISA assay kit (ab176647; Abcam) following the manufacturer’s instructions.

### 2.6. Histology

#### 2.6.1. Hematoxylin and Eosin Staining

The tissue was fixed in 10% buffered formalin for 24 h and embedded in paraffin. Before the tissue sections were stained, they were dewaxed and dehydrated. For the dewaxing step, xylene was used, and for the dehydration step, decreasing concentrations of ethanol were used. Afterwards, the sections were rinsed in double-distilled water (ddH_2_O), stained in hematoxylin solution for 2 min, rinsed again in tap water and ddH_2_O, dehydrated, counterstained in eosin for 2 s, dehydrated, and cleaned with xylene (all reagents from Sigma Aldrich).

#### 2.6.2. Goblet Cells Staining

Following fixing in formalin and embedding in paraffin, sectioned tissue was dewaxed in xylene and dehydrated in decreasing concentrations of ethanol. Afterwards, the tissue was rinsed in ddH_2_O before 1% alcian blue solution was pipetted directly on the samples. The tissue sections were incubated in a moisture chamber for 30 min. Then, the samples were rinsed in tap water for 3 min and 10 s in ddH_2_O. Next, 1% periodic acid was pipetted on the tissue, incubated for 10 min in the moisture chamber, and rinsed again in tap water for 3 min and ddH_2_O for 10 s. Afterwards, Schiff’s reagent was pipetted on the tissue, incubated for 15 min, and rinsed again in tap water and ddH_2_O. After that, the tissue was counterstained with hematoxylin for 2 min, followed by rinsing in tap water for 20 s and ddH_2_O for 10 s. The last steps were dehydrating the tissue in ascending concentrations in EtOH and cleaning it with xylene for 4 min. All reagents were purchased from Sigma Aldrich.

### 2.7. Imaging

Images were taken using Axio Lab 5× HSF 22 microscope, Axiocam ERc 5s camera, and Zeiss Efficient Navigation (ZEN) software (Zeiss efficient Navigation) (Carl Zeiss Microscopy, Oberkochen, Germany). The images were analyzed using ImageJ software (version 1.53, NIH, Bethesda, MD, USA).

### 2.8. Metabolomics

#### 2.8.1. SCFA and MCFA Detection

Detection of SCFA and MCFA was carried out with a derivatization method using liquid chromatography-mass spectrometry (LC-MS). Shortly after, 10–15 μg frozen cecum content was homogenized in Precellys homogenizing tubes containing extraction buffer (methanol:chloroform:water; 2.5:1:0.5) and 1.4 mm ceramic beads. After vortexing and centrifuging the samples for 5 min at 10,000× *g* at 4 °C, 600 μL of the supernatant was transferred to a new Eppendorf tube and the extraction was repeated. One hundred microliters of the combined supernatant was dried in a SpeedVac concentrator for 60 min at 45 °C. Dried pellets were resuspended in 150 μL acetonitrile:water (1:1) and 40 μL 40 mM 2-NPH, 40 μL 250 mM EDC, and 40 μL 3% pyridine, and incubated for 30 min at 60 °C. Samples were dried a second time in the SpeedVac concentrator for 75 min at 45 °C. To avoid phase separation, samples were resuspended in 50 μL acetonitrile:water (1:1) and diluted with 50 μL of acetonitrile:water (9:1). Samples were kept at 10 °C before analysis by LC-MS using an Ultimate 3000 (Thermo Scientific) and a micrOTOF-Q II (Bruker Daltonics, Bremen, Germany) with an Atlantis T3 3 μm column (2.1 × 150 mm, Waters, Milford, MA, USA). Mobile phase B (acetonitrile) was increased from 5% (0–2.5 min) to 90% (8 min), followed by a 5 min hold at 90%, then the column was washed with 95% mobile phase A (H_2_O) for 2 min; both solvents contained 0.1% formic acid.

#### 2.8.2. Bile Acids Quantification

Extraction and measurement of BAs were carried out as described previously [23]. Briefly, after adding nine times the volume of methanol to the samples, approximately 7 mg of small intestinal samples were cut and homogenized in Precellys tubes with 1.4 mm ceramic beads. Supernatants were transferred to Eppendorf tubes and were centrifuged at 15,000× *g* at 4 °C for 10 min. After repeating the centrifugation step, supernatants were pipetted into HPLC vials and were analyzed by LC-MS in positive modus using an LCMS-8040 Liquid Chromatograph Mass Spectrometer (Shimadzu Corporation, Kyoto, Japan) with an Atlantis T3 3 μm column (2.1 × 150 mm, Waters). The column temperature was kept at 30 °C. Gradient started from 30% B (acetonitrile/methanol (3/1), *v*/*v*) and 70% A (water). After 5 min, B was increased to 100% within 20 min and kept constant for 20 min. Afterward, the mobile phase was set back to 30% B for 10 min. Both solvents contained 0.1% formic acid with 20 mM ammonium acetate.

#### 2.8.3. Untargeted Metabolomics

Plasma samples (25 µL) were treated with 150 µL of cooled acetonitrile. Following equilibration for 30 min on ice, the suspension was centrifuged (10 min, 12,000 rpm, 4 °C). The clear supernatant was transferred into 1.5 mL autosampler vials with 250 µL inserts and stored at −20 °C until further analysis.

UHPLC-TOF-MS/MS analysis was performed using an Exion LC UHPLC system (Sciex, Darmstadt, Germany) connected to a TripleTOF 6600 mass spectrometer (Sciex) using electrospray ionization (ESI) in positive and negative mode. The UHPLC systems consisted of two Exion LC AD pumps, an Exion LC degasser, an Exion LC AC column oven, an Exion LC AD autosampler, and an Exion LC controller. Chromatography was carried out using a RP18 stationary phase (Kinetex C18, 100 × 2 mm, particle size 1.7 μm; Phenomenex, Aschaffenburg, Germany) using a constant flow rate of 0.25 mL/min at a column temperature of 40 °C. The mobile phase consisted of (eluent A) water and (eluent B) acetonitrile, both containing 0.1% formic acid. The gradient elution started with 5% B for 2 min and increased to 100% B in 11 min, held for 2.5 min, decreased to the initial ratio of 5% B within 0.5 min, and followed by 4 min of re-equilibration. The injection volume of all samples was 1 μL, and the autosampler was maintained at 10 °C. The mass spectrometer was operated in the Information Dependent Acquisition mode (IDA) for fragment spectra measurement. After starting with a high-resolution scan of the intact precursor ions from 50 to 1000 *m*/*z* for 250 ms, fragment ions were generated by means of collision-induced fragmentation for the eight most abundant precursor ions per cycle. The resulting fragment spectra were recorded in the high sensitivity mode between 50 and 1000 *m*/*z* (50 ms acquisition per experiment). Ion spray voltage was set at −4500 V in negative and 5500 V in positive mode. The following source parameters were applied: curtain gas 35 psi, gas one with 55 psi, and gas two with 65 psi at temperature 500 °C. Declustering potential was set to 80 V for all experiments, while the collision energy was 10 V for precursor ion scans and 35 V, including 20 V collision energy spread for the fragment ions. For normalization of longitudinal shifts during data acquisition, a quality check (QC) sample was prepared by mixing aliquots (10 µL) of each sample. The combined QC sample was injected into the LC-MS system after every five samples, while samples were analyzed after randomization for the avoidance of batch effects.

Raw mass spectrometry data was converted to the mzML file format using Proteowizard [26] and subsequently processed using the xcms package (version 3.18.0, open source software available at Bioconductor [27]) within the statistical programming environment R (version 4.2.1; open source software available at https://cran.r-project.org, accessed on 25 July 2022). Peak areas of detected features in the individual samples were used for the discriminative analysis of differences between experimental groups by Partial Least Squares—Discriminant Analysis (PLS-DA) with seven-fold cross-validation and 100 repeats to reduce the impact of the randomly allocated folds during each repeat (ropls package, version 1.28.2; open source software available at Bioconductor [28]).

Identification of bile acid features in the untargeted metabolomics data was performed by comparison of fragment mass spectra and retention times to an in-house library of primary and secondary bile acids [29,30].

### 2.9. Statistics

Statistical differences between the groups were assessed using one-way ANOVA and post-hoc analysis following Tukey’s method; *p* < 0.01 was considered statistically significant. Furthermore, *p* < 0.1 but not statistically significant considering correction for multiple testing was displayed on the figures and frequently referred to in the text as a trend.

## 3. Results

### 3.1. All Restrictive Diets Modulate the Expression of Inflammatory and Antibacterial Factors

In order to compare the impact of selected restrictive diets on the intestine, five groups of eight mice were randomly assigned one of the diets of interest: CR, IF, FMD, KD, and control diet. The control mice were fed standard chow *ad libitum*. Animals from all diet groups involving energy restriction lost body weight, with the CR group having the lowest body weight on the last day of the experiment (Figure 1A). Correspondingly, CR, IF, and FMD mice showed decreased epididymal white adipose tissue (eWAT) mass with the greatest change for CR (Appendix A). KD did not affect body or eWAT weight. Following on our previous report showing that CR is associated with the downregulation of inflammatory genes in the mucosa along the GI tract [11], we first compared the impact of the diets on the expression of Stat1, Irf1, and TNFα in the mucosa of the jejunum and proximal colon. In general, each of the restrictive diets showed a statistically significant impact or a trend to decrease the expression of Stat1 and Irf1 in the proximal colon (Figure 1C). The mRNA expression in the jejunum (Figure 1B) and proximal colon (Figure 1C), as well as protein level in the jejunum (Figure 1D) for TNFα, was not affected by any of the diets. To further explore the potential of the diets in modulating the immune status in the GI tract, the protein levels of total NFκB and its phosphorylated form (p-NFκB (Ser536)) were quantified in the jejunum mucosa (Figure 1E,F). Although no statistically significant impact has been observed, KD, IF, and FMD resulted in a trend of increased levels of IKB, an inhibitor of NFκB [31] (Figure 1G,H). Similar to the jejunum, in the proximal colon, the level of p-NFκB (Ser536) was not affected by any of the diets (Figure 1I,J). Protein levels of cytokines and inflammatory factors were assessed in the epithelium of the jejunum; however, no changes were found in the levels of IFNɣ, TGFβ1, MIP3α, and IL17F, regardless of the diet (Appendix A). Further, the expressions of the cytokines and inflammatory factors were assessed in the mesenteric lymph nodes and Peyer’s patches. Surprisingly, Stat1 gene expression was increased in Peyer’s patches of IF and FMD mice (Figure 1K), while the expression of Irf1 matched the results in the jejunum and colon mucosa. Among the diets, FMD and IF showed the strongest impact on the expression of tested genes in Peyer’s patches. Further, the expression of the corresponding set of genes was analyzed in the superior mesenteric lymph nodes to picture the impact of immune factors circulating in the vessels surrounding the GI tract (Figure 1L). In general, the impact of the treatments was much milder in the lymph nodes compared to the intestinal mucosa and Peyer’s patches, and it was comparable between the diets.

To assess immune status in the GI tract, the abundance of CD45+ cells was measured in Peyer’s patches and mesenteric lymph nodes. A stronger impact was observed for Peyer’s patches, where KD triggered a decrease in the percentage of the cells, and CR showed a similar tendency (Appendix A). The percentage of the cells was not statistically significantly affected in mesenteric lymph nodes of any of the diet groups (Appendix A).

### 3.2. Microbiota Composition Changes Depend on Energy and Macronutrients Restriction

Seeing that each of the diets can affect the immune status and suspecting consequent impact on gut microbiota, the gene expression of antimicrobial factors was analyzed. Oas1a and Nod2 mRNA expression was statistically significantly downregulated in the jejunum mucosa of all of the diet groups compared to the control (Figure 2A). Additionally, the expression of MyD88, Reg3γ, and Tlr3 genes was also decreased, although not all the differences were statistically significant. In general, KD consistently showed the strongest impact on gene expression. Only KD tended to reduce the expression of Tlr3, while Rsad2 was not affected by any of the diets (Figure 2A). A similar effect with additional downregulation of Rasd2 and a greater impact on Tlr3 was observed in Peyer’s patches (Figure 2B). In this tissue, KD and FMD seemed to have the most pronounced effect. The impact of the diets was much milder in the mucosa of proximal colon and mesenteric lymph nodes (Figure 2C,D). Surprisingly, and contrary to jejunum and Peyer’s patches, the expression of Tlr3 and Nod2 was increased by IF and FMD in the proximal colon (Figure 2C). In mesenteric lymph nodes, the expression of MyD88 showed a strong decreasing trend for all of the diets, and there was no consistent impact of the treatments on any of the other tested genes (Figure 2D). Recognizing that all of the tested restrictive diets regulate the synthesis of antimicrobial factors in the jejunum and Peyer’s patches on the gene expression level, the consequences of these regulations were verified by analysis of the relative abundance of selected bacteria in mice feces. The bacteria types were chosen based on our previous observations of microbiota composition changes during CR [11,22]. There were no differences in the abundance of phyla *Firmicutes* and *Bacteroides* in the feces of *ad libitum*-fed compared to the diet groups (Figure 3A, Appendix A). However, KD and CR groups tended to differ from IF and FMD groups (Figure 3A). Despite a similar effect on the expression of antimicrobial genes, for all of the tested diets, KD had opposing effects on the abundance of several bacteria compared to the other restrictive diets. CR and IF increased relative *Lactobacillus* levels, while KD depleted them (Figure 3B). A corresponding contrasting trend between CR and KD was observed for *Parasuttella* (Figure 3C). CR, IF, and FMD diminished *Deferribacteres* levels, but KD raised it (Figure 3D). Of the tested bacteria, only *Akkermansia* was consistently stimulated by each diet; however, it was not statistically significant (Figure 3E).

### 3.3. Metabolites Composition Changes Depend on Energy and Macronutrients Restriction

Considering the role of microbiota in SCFA and medium-chain fatty acids (MCFA) synthesis, the levels of the fatty acids were measured in the cecum content of the animals submitted to the dietary treatments. CR, IF, and FMD consistently decreased the levels of acetate (C2), butyrate (C4), and caproate (C6) cecal fatty acids (Figure 3F,G). KD resulted in the reduction of C4 to a similar extent as CR, IF, and FMD, but it did not affect C2, propionate (C3), or valerate (C5). Concerning MCFA, CR, IF, and FMD reduced the levels of cecal C6, FMD increased caprylate (C8) level, while IF and FMD showed a trend towards reduction of laurate (C12). Caprate (C10) was detected only in the cecum of KD mice (Appendix AH). Crucially, KD was supplemented with MCFA; therefore, its levels were very high in the cecum (Appendix A). Interestingly, cecum size was decreased in KD-fed mice, while energy-restricted mice tended to have enlarged cecum (Figure 3H).

The evinced mutual regulation involving gut microbiota and BAs [32,33] prompted us to analyze the levels of BAs in the epithelium of the ileum. Increased levels of UDCA, DCA, CDCA, and CA were found for CR, IF, and FMD, as well as TDCA for IF and FMD (Figure 3I–M). Whereas KD did not trigger changes in the level of any of the tested BAs. A similar trend was observed for BAs UDCA, DCA, and CA in the plasma (Figure 3N–P).

Further analysis of plasma metabolites indicated clear metabolites clustering with a distinctive profile for KD and control diet, while IF, CR, and FMD were most alike (Figure 4A,B).

### 3.4. Restrictive Diets Affect Mucin Production and Intestine Morphology

As microbiota and interleukins are some of the major factors contributing to goblet cell differentiation and mucus production, we performed staining of the colon (Figure 5A, replicates in Appendix A) and small intestine (Figure 5B). The diet treatments, particularly CR and IF, seemed to moderately increase the intensity of the staining of the colon sections. Similarly, in the small intestine, the diets resulted in a modest increase in the number of goblet cells; however, the difference was not statistically significant (Figure 5C). The mRNA expression of Muc13 in jejunum mucosa was higher in CR, IF, and FMD groups compared to *ad libitum*-fed mice (Figure 5D). On the contrary, Muc2 gene expression was not statistically significantly affected (Figure 5D). Also, the diets did not change the expression of Muc2 or Muc13 genes in the proximal colon mucosa (Figure 5E). Overall, the results suggest possible modest changes in mucus production in the GI tract of mice submitted to CR, IF, and FMD.

We then further investigated morphological changes triggered by the diets. KD resulted in the accumulation of lipid droplets within the villi (Figure 5F, Appendix A). In the small intestine, KD increased the villi length, while CR and IF showed a similar trend (Figure 5F,G). Also, KD and CR showed a trend towards reduced crypt depth (Figure 5H) and KD towards reduced intestinal *muscularis externa* thickness (Figure 5I). The changes were accompanied by decreased levels of zonulin (ZO-1) mRNA, while occludin gene expression was not affected in the mucosa of the jejunum (Figure 6A). Interestingly, the protein level of 65 kDa occludin was not changed during the diet treatment (Figure 6B), but KD triggered the occurrence of a shorter 50 kDa form of occludin (Figure 6C,D). In the colon, the protein and mRNA expression of occludin and ZO-1 was not changed upon diet exposure (Figure 6E–H).

### 3.5. Modulation of Autophagy, Mitochondria, and Oxidative Stress by Restrictive Diets

The beneficial outcomes of CR and IF are partly due to stimulation of autophagy, impact on mitochondrial function, and reduction of oxidative stress [34,35]. Therefore, we assessed the expression of markers of these features. First, there was no statistically significant impact of the diets on ATG7 protein expression (Figure 7A,B), whereas all of the diets tended to stimulate LC3 protein; however, only the impact of KD was statistically significant (Figure 7C,D). No impact on Atg7, Atg12, or Lc3 mRNA levels was measured in the jejunum mucosa (Figure 7E). In the mucosa of the proximal colon, the protein levels of ATG7 and LC3 were not affected (Figure 7F–I), although there was a trend towards increased expression of ATG7. The expression of the Atg7 gene was increased by FMD, CR, and nearly statistically significantly by KD (Figure 7J). The levels of Atg12 and Lc3 mRNA were not affected. Further, mitochondrial DNA levels in the mucosa of the jejunum tended to be increased by KD and CR (Figure 7K). Ucp2 and Tfam mRNA expression in the intestinal mucosa were not affected by the diets (Figure 7L). Whereas TOMM20 protein levels were stimulated by KD and IF (Figure 7M,N). In the mucosa of the proximal colon, TOMM20 and Ucp2 expression was not changed, whereas Tfam mRNA levels were reduced in KD and similarly tended to be decreased in CR and IF mice (Figure 7O–R). Finally, the expression of anti-oxidative enzymes was assessed. The levels of MnSOD mRNA were decreased by most of the diets in the mucosa of the jejunum (Figure 7S). Trx2 showed a similar trend, whereas the gene expression of glutathione transferases (Mgst1, Gsta3) tended to be elevated. The gene expression pattern for MnSOD was similar in the mucosa of the jejunum and proximal colon (Figure 7T). The expression of Trx2 was not affected in the colon, and only FMD modified the expression of Catalase, Mgst1, and Gsta3.

### 3.6. The Impact of Restrictive Diets on the Markers of Energy Limitation

Next, the factors previously found to be associated with energy restriction, including serine-threonine kinase AKT, metallothionein 2 (Mt2), and peroxisome proliferator-activated receptor gamma coactivator 1- α (PGC-1α) α were assessed. The dietary restrictions resulted in an increase in phosphorylated AKT levels in the jejunum mucosa of FMD-fed mice and a similar trend was observed for the other diets (Figure 8A,B). KD, CR, and FMD had the strongest stimulating impact on Mt2 and Pgc1α mRNA expression in the jejunum (Figure 8C). In the colon, the effect was much less pronounced with p-AKT not being affected, and statistically significant results were observed only for KD on Mt2 mRNA (Figure 8D–F).

## 4. Discussion

This work presents a comprehensive study comparing the impact of restrictive diets on the gut in the context of inflammation, microbiota, gut morphology, autophagy, mitochondria, and restriction-specific parameters. In general, the applied diets are very similar in terms of reduction of the expression of inflammatory genes and proteins; antimicrobial genes; mucus production; gut morphology; and autophagy-, mitochondria-, and restriction-related factors. However, they differ in terms of their effect on microbiota and BAs composition as well as the occurrence of occludin 55 kDa. Scrutinizing the magnitude of the impact of the diets, FMD and KD most consistently had the strongest effect. Notably, the outcomes were observed in several sections of the GI tract. Noteworthy, the impact on the jejunum was much more pronounced compared to the proximal colon.

Considering similarities and differences of the studied diets, they can be grouped by carbohydrates restriction (KD and FMD vs. CR and IF), ketogenic properties (with KD having the strongest), temporal restrain (IF and FMD), or the total energy restriction (CR and FMD vs. IF and KD). The common pathway strongly affected by all of the diets is insulin signaling. Therefore, with this study comparison between the impact of reduced insulin signaling and ketones signaling versus energy depletion may be claimed. However, one also needs to bear in mind the impact of the consumed nutrients. The composition of KD strongly differed from chow given to CR, IF, and FMD mice. Importantly, FMD received highly nutritious supplements during the restriction cycles. Further, KD is fed *ad libitum* while CR, IF, and FMD diets are associated with reduced energy intake and hunger. Taking into account this essential difference compared to the other restrictive diets as well as the range of performed assays, the overall outcomes and scale of changes were unexpectedly similar between all the diets.

Our study analyzed the outcomes of the relatively new but intensively researched dietary protocol FMD. FMD ameliorates the symptoms and pathogenesis of IBD [14,15]; however, the diet has never been investigated in the intestine in the context of basic, non-diseased conditions. We show that FMD impacts similar parameters as other restrictive diets. However, it has a somewhat stronger impact on the protein levels of p-AKT, SCFA cecum levels, and anti-oxidative and antimicrobial gene expression. Considering that fecal transplant from FMD mice contributes to neuroprotection in Parkinson’s disease mouse model [18], the here-reported similarities in the gut microbiota of FMD, IF, and CR mice hint at corresponding therapeutic properties of IF and CR. Overall, our results imply that FMD can be applied as a preventive measure to maintain low-immune reactivity, improve gut microbiota and bacteria metabolites composition, as well as manage oxidative stress.

KD has been applied for over 100 years to reduce the occurrence of epileptic seizures; however, there is a limited amount of information concerning its impact on the GI tract. In our study, KD was distinguished from the other applied diets by its impact on the microbiota and BAs composition, as well as the occurrence of 50 kDa occludin. The truncated 50-kDa occludin form is likely not a distinct translated product but a proteolytic fragment whose presence is associated with increased activity of metalloproteinases and disruption of the overall tight junction complex [36,37]. Correspondingly, low levels of tight-junctional resistance correlate positively with the 50-kDa form and negatively with the 65-kDa form in human cervical epithelial CaSki cells [38]. Notably, in the intestine, the role of different isoforms remains unexplored.

Previously, we showed that CR reduces the expression of inflammatory genes along the intestinal tract [11,22]. Several restrictive diets have been previously shown to extinguish inflammation and attenuate autoimmune diseases. Importantly, inflammation also serves as an important marker of regeneration [14]. Here, we show that all the applied restrictive diets have similar properties in terms of down- and also upregulation of the gene expression of inflammatory factors in all tested tissues. Further, we show that the critical master regulator of inflammatory genes expression NFκB may be involved in mediating the effects of the diets. The magnitude of the inflammatory gene expression regulation was comparable in the mucosa of the small intestine and colon as well as in Peyer’s patches. Whereas the response in the mesenteric lymph nodes was much milder, indicating the strong GI-tract-centered impact of the diets. Likely, the here-presented mechanisms contribute to preventing IBS and gut inflammation, maladies for which therapeutic properties of restrictive diets have been reported in the past [6,9,13,14,15].

Previously, we showed that CR modulates the composition of intestinal and fecal bacteria [11,22]. Surprisingly, KD, which is similar to other restrictive diets in terms of anti-inflammatory and antimicrobial gene and protein expression, showed the opposite impact on the abundance of several types of bacteria. The difference between KD and other restrictive diets confirms that the composition of consumed nutrients that further feed and breed bacteria is the main factor shaping gut bacteria [39]. It is also essential to consider that the beneficial impact of the diets may be partly bacteria-independent or rely on other bacteria whose abundance was not measured within this study.

The intestinal mucus layer is part of the innate mucosal intestinal barrier that reduces antigen and bacteria exposure to the epithelium, protecting and contributing to maintaining intestinal homeostasis [40]. Fasting changes mucus thickness already after 48 h [41]; interleukins tune goblet cells differentiation [42]; NFκB [43] and BAs [44] increase mucus production; while SCFAs stimulate mucus release [45]. Therefore, multiple factors assessed in this study impact the mucus layer. During food restriction, gut bacteria become dependent on mucin glycans for energy [46]. Furthermore, the gut microbiota is fundamental for goblet cells’ differentiation, function, and the formation of a proper mucus layer [47], and *Lactobacillus* species are known as mucus production stimulators [48]. Correspondingly, the observed changes in *Lactobacillus* abundance corresponded to Muc13 gene expression. However, here-reported changes in mucus-related parameters under restrictive diet conditions are very modest in both the small intestine and colon.

Restrictive diets trigger a rise in the ratio of villi length to crypt depth, which likely aims at increasing surface area for nutrient absorption. It is a sound response to energy restriction; however, the strongest difference was observed for *ad libitum* consumed KD. Interestingly, ketone bodies signal to stem cells, altering differentiation and regeneration [49]. Whereas medium-chain triglycerides, known for their ketogenic properties [50], are also applied to increase villi length and, therefore, enhance nutrient absorption and body weight gain [51].

Autophagy is induced to generate energy that supports metabolic processes and is often claimed to be one of the main benefits of fasting and caloric restriction [35]. At the same time, it is a key process required for a functional GI tract in terms of maintaining intestinal barrier integrity, anti-microbial defense, mucosal immune response [52], enteroendocrine cells’ function [53], mucin formation, and secretion by goblet cells [54]. Results of our study imply that restrictive diets stimulate autophagy in the GI tract, although the impact we measured is relatively mild.

Mitochondrial functionality and cellular energy metabolism are regulated by gut bacteria and are associated with cancer, celiac, and Crohn’s disease [55,56,57]. KD has previously been shown to rescue IBD-associated mitochondrial dysfunction in the rat intestine [58], increase mitochondria turnover [59] and mitochondrial protein acetylation [60]. Similarly, CR exhorts its beneficial effect on the intestine via mitochondria [61]. In our study, the KD and CR showed a tendency towards upregulation of mtDNA and TOMM20 in jejunum but did not impact the levels of Ucp2 and tended to reduce the expression of mitochondrial transcription factor Tfam in the colon, which may indicate changes in the mitochondrial turnover compared to its activity. Whereas several diets in our and previous studies [62] tend to increase the expression of Pgc1α, which is crucial for mitochondria function, the results may also be interpreted as contradictory, which fits well with confusion concerning the stimulatory role of CR in preserving mitochondrial function without increasing biogenesis [63] as well as enhancing mitochondria biogenesis [64] or stimulating mitoptosis [35]. Importantly, in the light of the complex interplay between mitochondrial function, mitochondrial biogenesis, mitophagy versus mitochondria rejuvenation, as well as the number of mitochondria per cell versus mitochondrial activity, they all lack selective markers that allow distinguishing between these processes, which leads to a farfetched interpretation of the experimental results. Further, mitochondria function not only pertains to energy generation but also plays an essential role in apoptosis and the generation of reactive oxygen species. Therefore, based on our results, KD and CR may affect mitochondria in the intestinal epithelium; however, the nature and outcome of this regulation need further investigation.

For the here-presented experimental setup, the most commonly occurring versions of each diets were chosen. However, the length of the treatment, as well as the extent of restriction or nutrient composition varies strongly among different types of each diet applied in real-life and experimental settings. CR is experimentally applied within the range of 10–50% reduction in food intake. KD has multiple versions with various types and content of fat, which is vital to consider as, e.g., medium chain triglycerides (MCTs) exert greater anti-inflammatory effects in inflammatory gut models and high fat diet-obese mice [65,66]. However, in the skin, the effect was the opposite [67]. Each diet can be applied for several days up to life-long. Therefore, it is crucial to underline that the obtained results are specific to the selected experimental conditions. Future experiments exploring differences between subtypes of the diets may broaden the choice of applicable dietary conditions. Moreover, KD outcomes demonstrated that carbohydrate restriction has beneficial outcomes. However, previously, also protein restriction has been used as therapeutic means. This leads to a question of what other types of restrictions could have similar effects and why such a broad range of restrictions have a beneficial impact.

We conclude that despite drastic differences in the dietary protocols, the outcomes strongly align. Each of the diets is likely beneficial when applied as a drug-free approach to maintaining GI tract health. Notably, the beneficial impact of the diets can be achieved via KD without restricting energy intake, boosting hunger, or reducing basal metabolic rate, which often accompanies CR and IF. On the other hand, strict excision of carbohydrates may be more challenging to adhere to for some individuals than mild energy restriction, limiting eating window, or periodic FMD. Considering outcomes in the basic situation, it will be interesting to compare the diets in the context of inflammatory gut diseases. Finally, verification of the observed phenotype in a female model is necessary.

## Figures and Tables

**Figure 1 nutrients-14-03120-f001:**
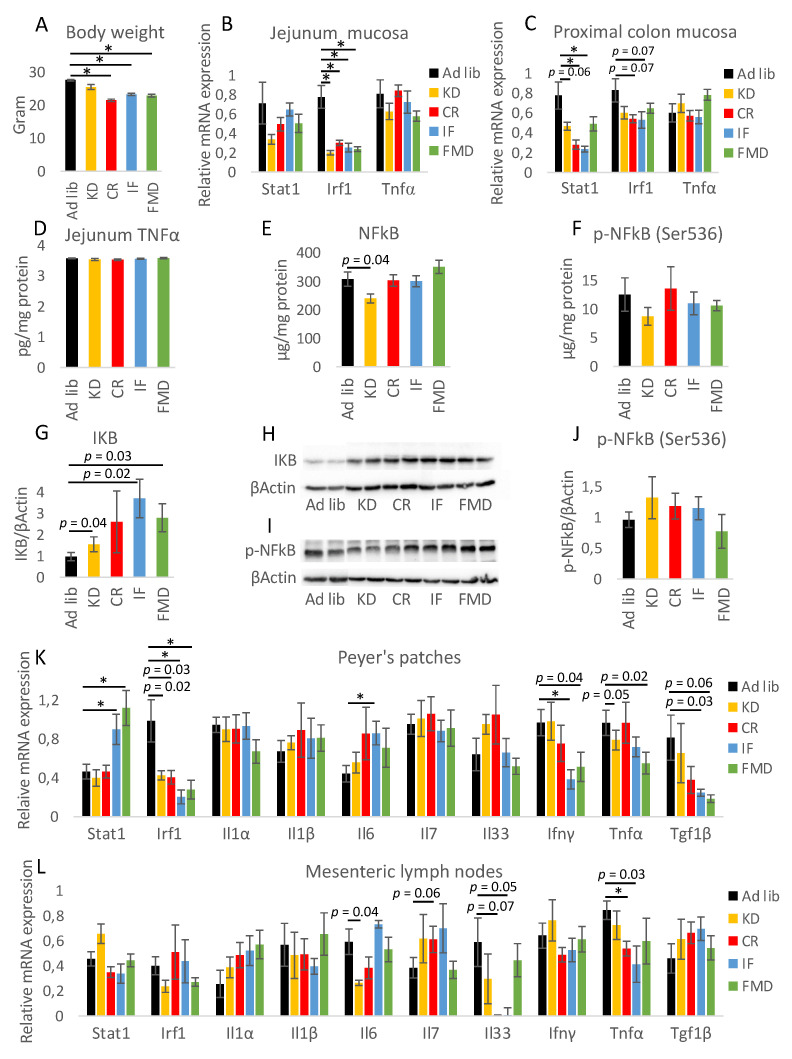
Restrictive diets reduce the expression of inflammatory factors in the gastrointestinal (GI) tract. Bodyweight was verified on the last day of the experiment (**A**). Gene expression of inflammation-related genes was measured in the mucosa of the jejunum (**B**) and proximal colon (**C**). The protein level of TNFα (**D**), NFκB (**E**), phosphorylated NFκB (**F**,**I**,**J**), and IKB (**G**,**H**) was measured in the mucosa of the jejunum (**D**–**H**) and colon (**I**,**J**). The quantification of Western blot-based protein expression measurement (**H**,**I**) represents an average of 6–8 samples per experimental group. The mRNA expression of inflammatory factors was measured in Peyer’s patches (**K**) and mesenteric lymph nodes (**L**) using qRT-PCR. Statistical significance between experimental groups was evaluated using ANOVA followed by Tukey’s post hoc test. * *p* < 0.01; *n* = 6–8. Data are presented as the mean ± SEM.

**Figure 2 nutrients-14-03120-f002:**
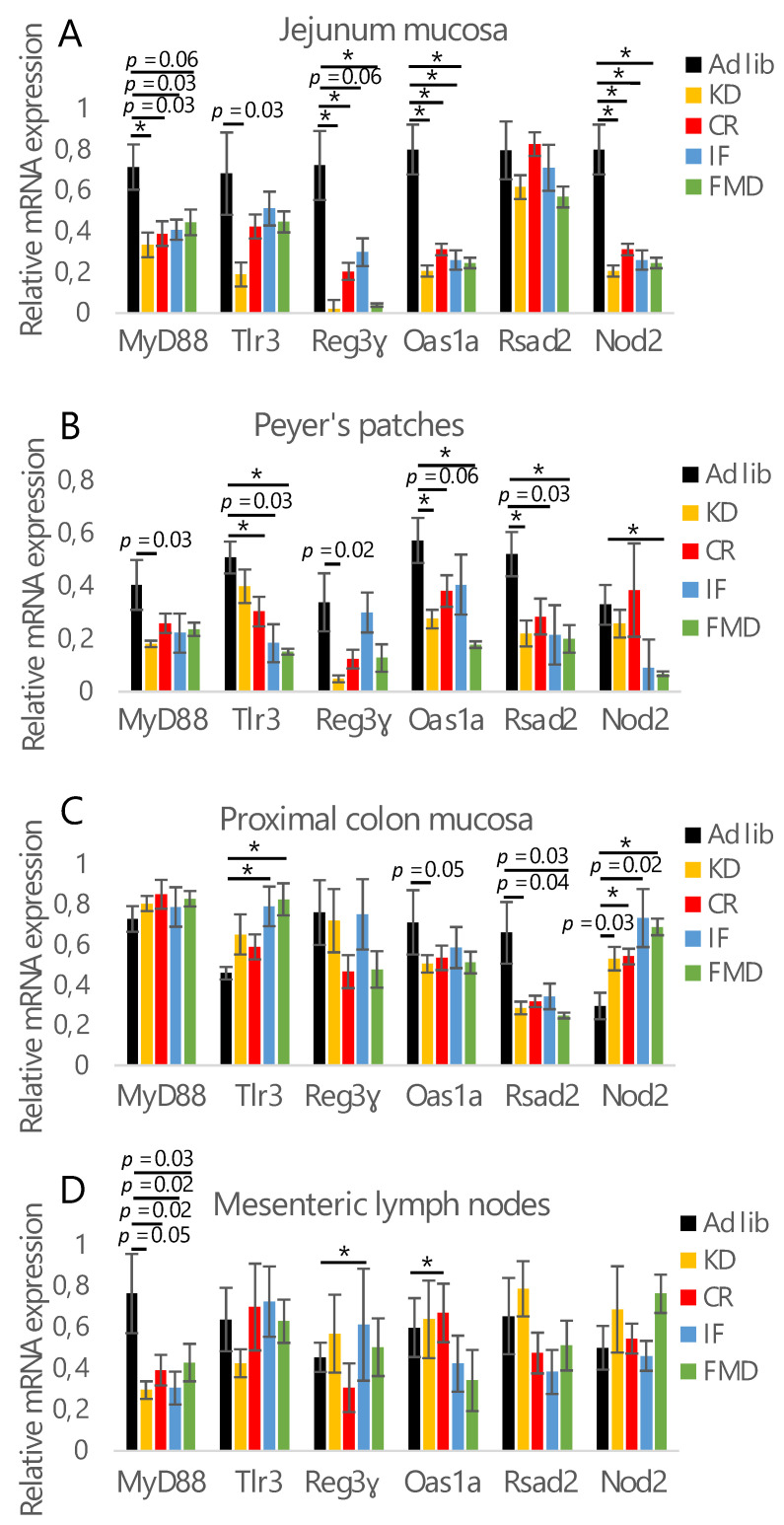
Restrictive diets reduce the expression of antimicrobial factors. The mRNA expression of microbiota-related factors was measured in the mucosa of the jejunum (**A**), Peyer’s patches (**B**), proximal colon mucosa (**C**), and mesenteric lymph nodes (**D**). The groups were compared using ANOVA followed up by Tukey’s post-hoc test. * *p* < 0.01; *n* = 6–8. Error bars indicate ± SEM.

**Figure 3 nutrients-14-03120-f003:**
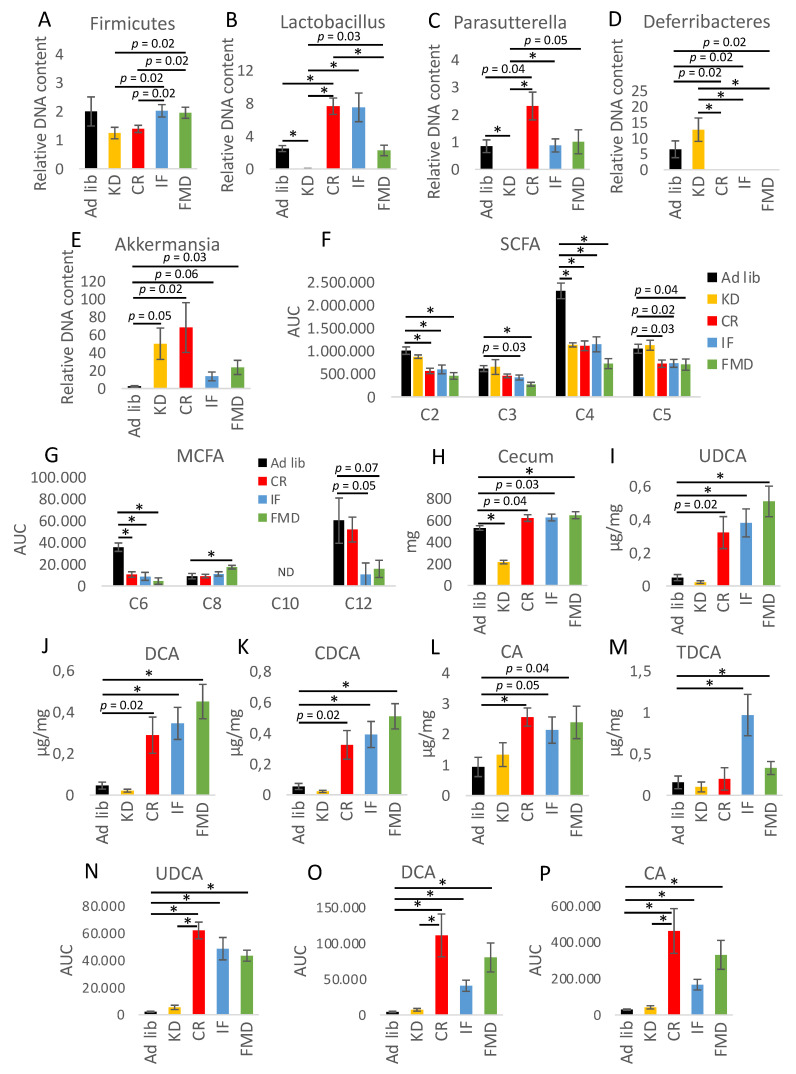
Restrictive diets modulate microbiota composition, the levels of microbiota-related factors, and metabolites. The relative abundance of Firmicutes (**A**), Lactobacillus (**B**), Parasutterella (**C**), Deferribacteres (**D**), and Akkermansia (**E**) in mice feces was assessed with qRT-PCR. The levels of short-chain fatty acids (SCFA) (**F**) and medium-chain fatty acids (MCFA) (**G**) were measured in mice cecum using HPLC-MS/MS. Mice cecum weight was measured (**H**). The bile acids UDCA (**I**,**N**), DCA (**J**,**O**), CDCA (**K**), CA (**L**,**P**), and TDCA (**M**) content in the ileum mucosa (**I**–**M**) and plasma (**N**–**P**) were quantified. ND in panel G indicates not detected. The groups were compared using ANOVA followed up by Tukey’s post hoc test. * *p* < 0.01; *n* = 6–8. Error bars indicate ± SEM.

**Figure 4 nutrients-14-03120-f004:**
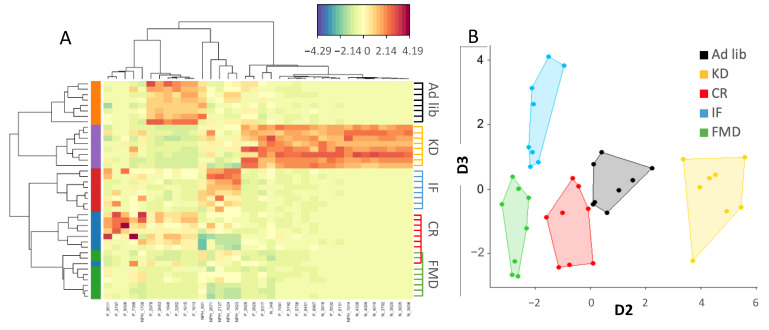
KD plasma metabolome composition is distinct from other restrictive diets. Plasma metabolome indicated clustering of metabolites specific to each diet (**A**), and PLS analysis (**B**) confirmed distinct features separating the diets.

**Figure 5 nutrients-14-03120-f005:**
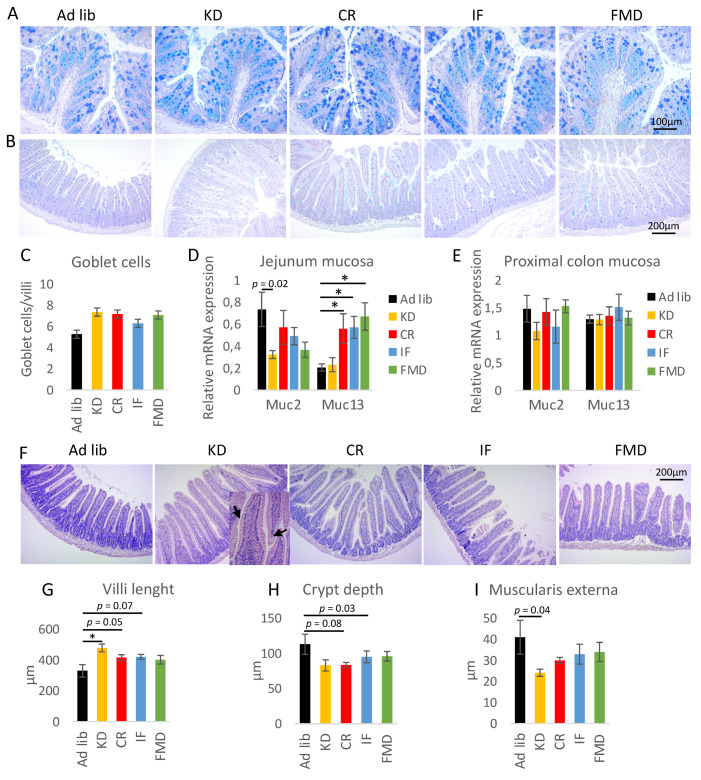
Restrictive diets have a small impact on goblet cells and mucus production in the GI tract, but affect intestine morphology. Histology sections of the colon (**A**) and jejunum (**B**) were stained with alcian blue periodic acid-Shiff (AB-PAS) to visualize goblet cells and mucus. The number of goblet cells per villi was counted in the jejunum sections (**C**). The mRNA expression of mucin (Muc2 and Muc13) genes was quantified in the mucosa of the jejunum (**D**) and proximal colon (**E**). Histological sections of jejunum were stained with hematoxylin-eosin (**F**), and the length of villi (**G**), depth of crypts (**H**), and thickness of muscularis externa (**I**) were quantified using ImageJ. Microscopy images were taken using Zeiss Axio Lab 5x HSF 22 microscope, Axiocam ERc 5s camera, and ZEN software (Zeiss efficient Navigation). The images were analyzed using ImageJ software. Statistical significance between experimental groups was evaluated using ANOVA and Tukey’s post hoc test. * *p* < 0.01; *n* = 6–8. Data are presented as the mean ± SEM. The scale bar in the last picture of A, B, and F is applied to all pictures.

**Figure 6 nutrients-14-03120-f006:**
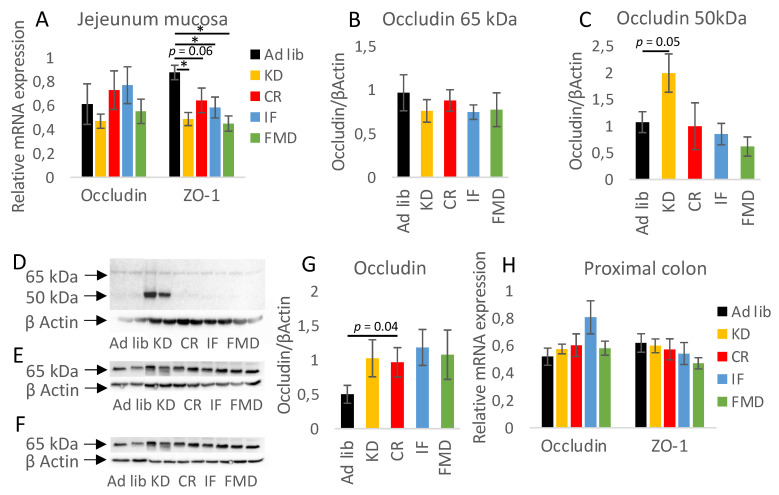
KD induces the occurrence of the truncated form of occludin. The relative mRNA level of zonulin (ZO-1) was measured in the mucosa of the jejunum (**A**) and proximal colon (**H**). The expression of occludin was quantified on the mRNA (**A**,**H**) and protein (**B**–**G**) levels in the mucosa of the jejunum (**A**–**D**) and proximal colon (**E**–**G**). Statistical significance between experimental groups was evaluated using ANOVA and Tukey’s post hoc test. * *p* < 0.01; *n* = 6–8.

**Figure 7 nutrients-14-03120-f007:**
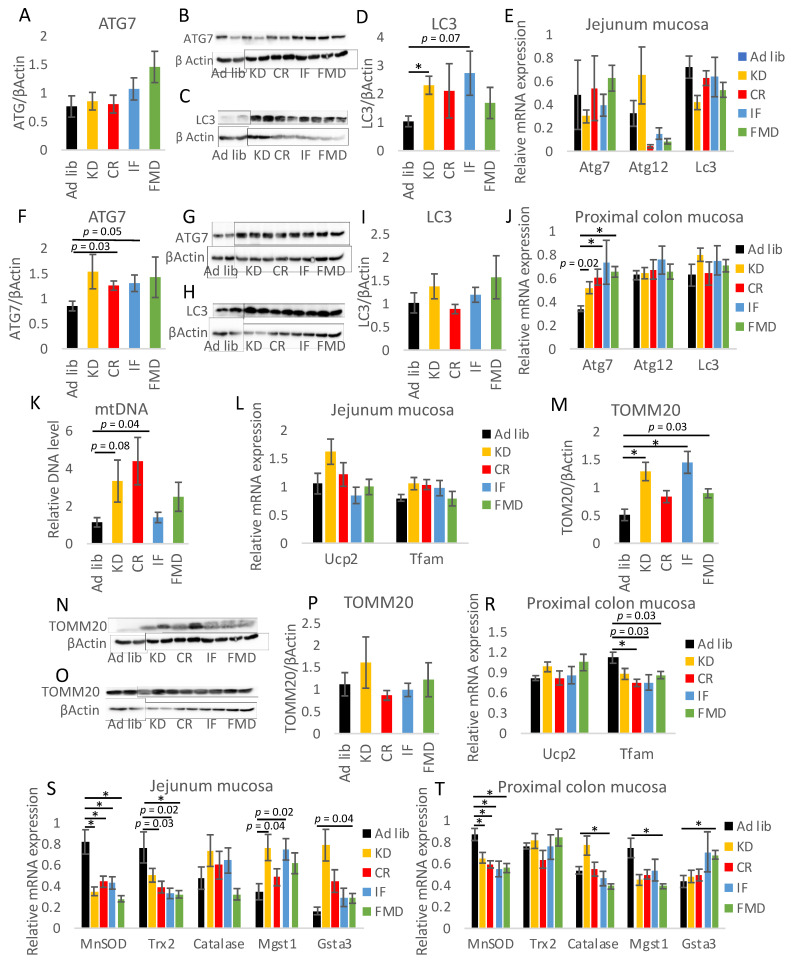
Restrictive diets affect the expression of autophagy- and mitochondria-related factors. Protein levels of ATG7 (**A**,**B**,**F**,**G**) and LC3 (**C**,**D**,**H**,**I**) were measured in the mucosa of the jejunum (**A**–**D**) and proximal colon (**F**–**I**). Gene mRNA expression of Atg7, Atg12, and Lc3 was measured in the mucosa of the jejunum (**E**) and the proximal colon (**J**). Mitochondrial DNA (mtDNA) level (**K**) was measured in the mucosa of the jejunum using qRT-PCR. Gene expression of Ucp2 and Tfam (**L**,**R**), as well as antioxidative factors (**S**,**T**), was quantified by applying qRT-PCR in the mucosa of the jejunum (**L**,**S**) and the proximal colon (**R**,**T**). The expression of TOMM20 (**M**–**P**) was quantified with Western blot in the mucosa of the jejunum (**M**,**N**) and the proximal colon (**O**,**P**). Statistical differences were assessed with ANOVA and by Tukey’s post hoc test. * *p* < 0.01; *n* = 6–8. Data are presented as the mean ± SEM.

**Figure 8 nutrients-14-03120-f008:**
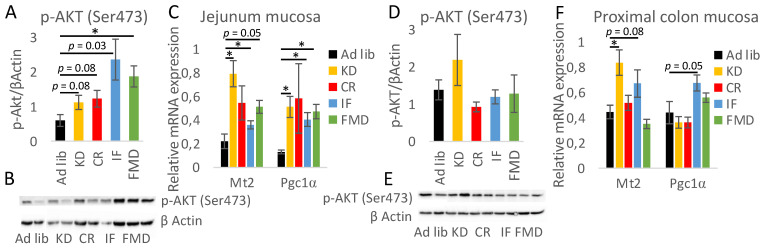
Restrictive diets affect the expression of CR-related factors. The protein level of phosphorylated AKT (p-AKT (Ser473)) was assessed in the mucosa of the jejunum (**A**,**B**) and proximal colon (**D**,**E**). Gene expression of Mt2 and Pgc1 was measured in the mucosa of the jejunum (**C**) and colon (**F**). Statistical differences were assessed with ANOVA and by Tukey’s post hoc test. * *p* < 0.01; *n* = 6–8. Data are presented as the mean ± SEM.

## Data Availability

All the data is presented in the article.

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
