# Peer review of "A Comparison of the Impact of Restrictive Diets on the Gastrointestinal Tract of Mice"

_nutrients, 2022, doi:10.3390/nu14153120_

Round 1

Reviewer 1 Report

The present work entitled: "A comparison of the impact of restrictive diets on the gastrointestinal tract of mice" gives an overview of the effects in cytokine production, metabolic changes, microbiota, and mucosa modification of different diets that restrict caloric intake or that change the macronutrient content. All diets impact the insulin pathway. It is noticeable that in this study the gut-associated tissue was studied, cytokine production was analyzed in the Peyer's patches and mesenteric lymph nodes. Thus, this is a very complete study of the metabolic and immune response to diverse diets.  As the author mention, the further step will be to determine the effect of these diets in a context of inflammation beyond the intestine.

Being said that I have some minor comments:

1. Can you please mention the age of the mice you used? How many mice di you used per group?

2. The cardiac puncture, mentioned in methods, was to draw blood?

3  In lines 175 and 173 you mean "moisture chamber"?

Author Response

  1. Can you please mention the age of the mice you used? How many mice di you used per group?

Thank you for this comment. The age and number of mice are now mentioned in the materials and methods section.

  1. The cardiac puncture, mentioned in methods, was to draw blood?

Yes, the cardiac puncture was used to draw blood. This information is now also added in the materials and methods section.

3  In lines 175 and 173 you mean "moisture chamber"?

Yes, thank you for noticing that. We have now corrected the description.

We would like to thank you for all your comments and help in improving the manuscript.

Reviewer 2 Report

In this manuscript, authors compared the impact of several restrictive diets on GI track. Several concerns regarding the methods and results need to be addressed:

Major:

Materials and Methods:

1. Please indicate the age and total number of mice at the initiation of the experiments in “2.1. Animal experiments”.

2. Statistics description is not clear. For example, Line 220-221, incomplete sentence and spelling errors “Statisticall differrences between the groups were assesed using one-way ANOVA and p-value lower than 0.05”??  Why would authors run student t-test rather than poc hoc analysis following one-way ANOVA? Was p<0.05 or p<0.01 considered significant? Authors may need consult a statistician for correct statistical analysis.

Results:

1. Is the bodyweight shown in Figure 1A the bodyweight gain or endpoint weight?  

2. Descriptions of results (or Figures) are not accurate. For example, Line 237-238, it stated that “ In general, each of the restrictive diets showed a statistically significant impact or a trend to decrease the expression of Stat1 and Irf1 in both tissues (Fig 1B-C)”.  However, none of the diets shows any impact on Stat1 in Jejunum mucosa as shown in Fig1B.

3.Some of the representative Figures are difficult to read.  For example, the lines used to indicate the differences (or comparisons) between groups are not well placed (Fig1A, G, K; Fig2A). It is difficult to read the comparisons without reading context.

Figure 3O, do the representative 65kDa Adlib blot and b-actin Ad lib blot are from the same protein samples (or same mice)? It seems that the ad lib blots are cut and pasted from a different blot. Please confirm and check this and other representative western blots in Figure 3 and 4.

Fig2J and L, symbol and abbreviation for each

Line 332-334, it stated that “The diet treatment, particularly CR and IF seemed to moderately increase the intensity of the staining of the colon sections suggesting an increased number of mucus-producing goblet cells”. Does the colon goblet cells counting support this claim? Please consider to quantify the goblet cells staining in colon sections and do comparisons.

Discussion.

Line 434-436, data presented in this MS did not indicate FMD has “stronger impact on the protein levels of p-NFκB”.  

This study examines the beneficial impacts of restrictive dietary regimen on GI track health in male mice. Will these diets exert the similar impacts in females?

Minor:

Grammatical/spelling errors

Line 379, Upc2.

Line 220-221, “Statisticall differrences between the groups were assesed………

Author Response

First of all, we would like to thank for the extensive revision, all your comments, and help in improving the manuscript. Please find below our answers to the raised points. 

Materials and Methods:

  1. Please indicate the age and total number of mice at the initiation of the experiments in “2.1. Animal experiments”.

The number and age at the beginning and end of the experiments have been indicated in the material and methods section 2.1.

  1. Statistics description is not clear. For example, Line 220-221, incomplete sentence and spelling errors “Statisticall differrences between the groups were assesed using one-way ANOVA and p-value lower than 0.05”??  Why would authors run student t-test rather than poc hoc analysis following one-way ANOVA? Was p<0.05 or p<0.01 considered significant? Authors may need consult a statistician for correct statistical analysis.

Thank you for noticing the oversight. We now corrected and unified the statistical method description in the whole text. The indicated results reflect the results of ANOVA with Tukey’s post hoc test. The description has been corrected in the materials and methods section as well as in the figures’ legends.

Results:

  1. Is the bodyweight shown in Figure 1A the bodyweight gain or endpoint weight?  

The presented data shows bodyweight at the endpoint. This information has now been provided in the results section and figure 1 legend.

  1. Descriptions of results (or Figures) are not accurate. For example, Line 237-238, it stated that “ In general, each of the restrictive diets showed a statistically significant impact or a trend to decrease the expression of Stat1 and Irf1 in both tissues (Fig 1B-C)”.  However, none of the diets shows any impact on Stat1 in Jejunum mucosa as shown in Fig1B.

We are sorry for this mistake. It has now been corrected.

3.Some of the representative Figures are difficult to read.  For example, the lines used to indicate the differences (or comparisons) between groups are not well placed (Fig1A, G, K; Fig2A). It is difficult to read the comparisons without reading context.

The figures have been modified. The lines indicating the results of the comparison have been adjusted.

Figure 3O, do the representative 65kDa Adlib blot and b-actin Ad lib blot are from the same protein samples (or same mice)? It seems that the ad lib blots are cut and pasted from a different blot. Please confirm and check this and other representative western blots in Figure 3 and 4.

In fact, the initial experiments included two types of KD, each with a different fat profile. However, there were no interesting differences between the two KD groups. Therefore, we have decided to show data for only one of them and, thus, we had to cut one KD group from each of the blots. Importantly, upon cutting the pictures, no visual modifications of brightness, contrast, or background correction have been applied that could influence the presented results and the actual differences between the groups.

The blots corresponding to the assed protein and b-actin come from the same animal (the same protein sample).

Fig2J and L, symbol and abbreviation for each

The missing references have been introduced.

Line 332-334, it stated that “The diet treatment, particularly CR and IF seemed to moderately increase the intensity of the staining of the colon sections suggesting an increased number of mucus-producing goblet cells”. Does the colon goblet cells counting support this claim? Please consider to quantify the goblet cells staining in colon sections and do comparisons.

Thank you for noticing this issue. We were previously considering goblet cells quantification; however, the quantification in the colon seems even more challenging than in the small intestine samples. In multiple instances, it is difficult to identify cells because of the strong background created by extracellular mucus. Also, contrary to the small intestine, in the colon, goblet cells strongly vary in size, which would also have to be accounted for and would create an issue with the arbitrary division between cell sizes. Our strategy was to provide in the supplementary figures more pictures showing differences between other replicates in order to make our point more convincing. If the quantification is necessary, we would ask for more than ten days of revision time.

Discussion.

Line 434-436, data presented in this MS did not indicate FMD has “stronger impact on the protein levels of p-NFκB”.  

We are sorry for this mistake. It has now been corrected.

This study examines the beneficial impacts of restrictive dietary regimen on GI track health in male mice. Will these diets exert the similar impacts in females?

Verification of the results in both sexes is an important aspect of any intervention. Unfortunately, we have not performed this verification yet. Therefore, we indicated the need for it in the discussion section.

Minor:

Grammatical/spelling errors

Line 379, Upc2.

Line 220-221, “Statisticall differrences between the groups were assesed………

Thank you. The spelling mistakes have been corrected.

Reviewer 3 Report

The manuscript by András Gregor et al. investigated the impact of restrictive diets on the gastrointestinal tract of mice. The study is of interest to the readers and I have the following suggestions:

1, The authors should further refine the figures to show the Y axis in each panel. I can hardly read the scale. 

2, The authors should use 16s sequencing to analyze the changes of the gut microbiota. 

3, Figures 2, 3, and 4 should be considered to be devided into two or more figures. I have never seen a figure cotains panels A to Z. 

Author Response

1, The authors should further refine the figures to show the Y axis in each panel. I can hardly read the scale. 

Thank you for your comment. The font size on the axis has been increased. Also, the spaces between the figures have been increased in most cases, making the figures easier to read.

2, The authors should use 16s sequencing to analyze the changes of the gut microbiota. 

We agree that it is a standard and useful approach. We regret not having performed the sequencing. However, having ten days for revision does not guarantee enough time to fulfill this request.

3, Figures 2, 3, and 4 should be considered to be devided into two or more figures. I have never seen a figure cotains panels A to Z. 

Thank you for this suggestion. The figures have now been split and modified for better readability.

We would like to thank for all your comments and help in improving the manuscript.

Round 2

Reviewer 3 Report

The authors have revised the manuscript accordingly. I suggest to accept this manuscript.